



# Internal climate variability and spatial temperature correlations during the past 2000 years

Pepijn Bakker[1], Hugues Goosse[2], and Didier Roche[1,3]

[1]Department of Earth Sciences, Vrije Universiteit Amsterdam, the Netherlands
[2]Earth and Life Institute (ELI), UCLouvain, Louvain-la-Neuve, Belgium
[3]Laboratoire des Sciences du Climat et de l'Environnement, LSCE/IPSL, Université Paris-Saclay, Gif-sur-Yvette, France

**Correspondence:** Pepijn Bakker (p.bakker@vu.nl)

**Abstract.** The spatio-temporal structure of natural climate variability has to be taken into account when unraveling observed climatic changes and simulate future climate change. However, based on the comparison of temperature reconstructions and climate model simulations covering the past two millenia, it has been argued that climate models are biased. They would simulate too little temporal temperature variability and too high correlations between temperature time series from different

continents. One of the proposed causes is the lack of internal climate variability in climate models on centennial time scales, for instance variability related to the Atlantic Meridional Overturning Circulation (AMOC).

We present a perturbed-parameter ensemble with the iLOVECLIM earth system model containing various levels of AMOC-related internal climate variability to investigate the effect on the spatio-temporal temperature variability structure. The model ensemble shows that indeed enhanced AMOC variability leads to more continental-scale temperature variability, but it also

increases the spatio-temporal temperature correlations between different continents. However, combining the iLOVECLIM results with CMIP5 model results and various PAGES-2K temperature field reconstructions, we find that neither model results or reconstructions are robust. We show overall agreement for the magnitude of continental temperature variability in models and reconstructions, but both the simulated and the reconstructed ranges are large. This is even more true when considering higher order metrices like inter-continental temperature correlations or temperature variability land-sea contrasts. For such metrics,

uncertainties in both model results and temperature reconstructions are so large that they hamper our ability to constrain simulated spatio-temporal structure of centennial temperature variability. As a result, we cannot determine the importance of AMOC variability for the climatic evolution over the past two millenia.

## 1 Introduction

Comparing reconstructed and simulated past climate variability helps us to understand natural climate variability, which is

important in the light of ongoing climate change (Braconnot et al., 2012; Deser et al., 2012). The most recent two millennia forms an important period in this respect because it i) is described by what is probably the highest density of palaeoclimate reconstructions of any past period (Ahmed et al., 2013; PAGES2k-Consortium, 2017), ii) is a period with relatively weak and well constrained external forcings allowing for a better investigation of unforced climate variability (Jungclaus et al., 2017) and iii) is a period which is very similar to the present and future climate in terms of mean climate, boundary conditions and





climate forcings except for human-induced changes (Schmidt, 2010).

Previous studies looking at temperature temporal variability have suggested that climate models simulate too little regional variability on multi-decadal and longer time scales (Laepple and Huybers, 2014; PAGES-2k-PMIP3, 2015). One can also compare reconstructed and simulated climate variability in the spatio-temporal domain, as the covariance varies as a function of the spatial and time scales inviestigated (Kunz and Laepple, 2021). For instance, PAGES-2k-PMIP3 (2015) compared the

PAGES-2K temperature reconstructions (Ahmed et al., 2013) for seven continental-scale regions with transient climate model simulations over the past 2000 years from the third Palaeoclimate Modelling Intercomparison Projects (PMIP3; Braconnot et al., 2012), and focused on the correlations between temperature time-series from different continental-scale regions. Based on this analysis, PAGES-2k-PMIP3 (2015) found that the reconstructions show weak surface air temperature covariance across continents, in contrast to relatively strong covariance found in climate models.

Several possible causes have been put forward to explain those model-data discrepencies, in both time domain and in the spatio-temporal domain. The model-data mismatches could be related to the reconstructions, with proxy-specific uncertainties inherent to the data that lower the reconstructed covariance (Hartl-Meier et al., 2017). Other sources of uncertainty are the model sensitivity to external forcings, such as volcanic eruptions and solar forcing, or the magnitude of the reconstructed external forcings. Interestingly, it has both been argued that the model response to external forcings is too weak, thus explaining the

lack of model variability on long time-scales (Laepple and Huybers, 2014), and that the model response to external forcings is too strong (Anchukaitis et al., 2010; Braconnot et al., 2012; PAGES-2k-PMIP3, 2015; Stoffel et al., 2015), thus explaining the large degree of temperature covariance in the models between different continents. Another explanation that has been forward to explain model-data mismatches in temperature variability is the lack in models of sufficient internal climate variability (Laepple and Huybers, 2014; Valdes, 2011). The idea behind the latter is that increased internal climate variability would in-

crease temperature variability and potentially weaken temperature covariance across continental-scale regions because it would add 'random noise' to the system. On the other hand, modes of internal climate variability have a clear spatial structure and can thus also enhance the spatio-temporal covariance by increasing the strength of climatic teleconnections between regions (PAGES-2k-PMIP3, 2015).

Here, we will investigate the impact of ocean-induced multi-decadal to multi-centennial climate variability on the spatio-

temporal covariance of the temperature evolution. Hereby, we test if increased internal climate variability can indeed improve the model-data comparison of temperature variability over the last 2000 years. We will specifically investigate the impact of ocean variability driven by the Atlantic Meridional Overturning Circulation (AMOC). To this end, we will present a perturbed-parameter ensemble of climate model simulations for the past two millenia that cover a range from very weak to very strong multi-decadal to multi-centennial AMOC variability. We will compare our model results with temperature reconstructions for

the past two millenia derived with different climate field reconstruction (CFR) methods (Neukom et al., 2019) and a subset of the CMIP5 last millennium simulations (Braconnot et al., 2012; Taylor et al., 2012). We will focus on two different spatial scales to investigate the inter-regional temperature coherency, namely inter-continental and between ocean and land. This will allow us to investigate the role of ocean-induced climate variability in altering spatial temperature variability coherency and to determine which level of ocean variability yields model results that are in best agreement with proxy-based temperature





reconstructions.

We limit our analysis of reconstructed temperatures for the past two millenia to the data set published by Neukom et al. (2019),
but many others exist (e.g. Ljungqvist et al., 2019; Zhang et al., 2018; Wang et al., 2017; Moreno-Chamarro et al., 2017; Franke
et al., 2016; Luterbacher et al., 2016). All these temperature reconstructions are affected by, among others, the spatial distri-
bution of the underlying records, their temporal extent, seasonality effects and the climate response of the various proxy types

(Anchukaitis and Smerdon, 2022). Given the goals of our study as outlined above, the temperature reconstructions of Neukom
et al. (2019) is the most suited as it covers sufficiently large spatial scales and includes an estimate of the methodological
uncertainties.

## 2  Methods

The work presented here is based on a perturbed-parameter ensemble of the *i*LOVECLIM Earth system model, on the PAGES-

2k temperature reconstructions (Neukom et al., 2019) and on a selected number of CMIP5 last millennium simulations.

### 2.1  *i*LOVECLIM perturbed-parameter ensemble

*i*LOVECLIM (here in version 1.1) is a code fork of the LOVECLIM 1.2 model (Goosse et al., 2010). It consists of a free
surface ocean general circulation model with an approximately 3°spatial resolution and 20 vertical layers. It is coupled to a
thermo-dynamical sea ice model and a quasi-geostrophic model atmospheric model, solved on a T21 spectral grid.


We performed a 70-member perturbed-parameter ensemble of pre-industrial control simulations with the *i*LOVECLIM
model. From this ensemble, we selected nine parameter sets that yield results that are both reasonable in comparison with
present-day observations (not shown) but also cover a large range of magnitudes in multi-decadal to multi-centennial AMOC-
related ocean variability. These nine parameter set include our default parameter set (experiment 1). Table A1 lists the values

that are used for the 10 perturbed parameters. Parameter selection was done using a Latin Hypercube sampling (LHS). The
ranges that are used in the LHS are given in Table A2. The choice of parameters to perturb is based on previous research by
(Loutre et al., 2011; Shi et al., 2019) and personal experience. Note that for some of the experiments (2-5), a limited number
of parameters was excluded from the LHS and instead we imposed parameter changes based on our knowledge of the relation
between those specific parameters and changes in the AMOC behavior in the model. This is specifically the case for the pa-

rameters controlling the size of the imposed precipitation correction between the Atlantic and the Pacific (Tables A1 and A2).
We assessed the degree of agreement with present-day observations for a selection of variables, namely the AMOC strength at
26°N (Moat et al., 2020), Northern Hemisphere (NH) and Southern Hemisphere (SH) sea-ice extent (Niederdrenk and Notz,
2018; Roche et al., 2012), the top-of-the-atmosphere radiative imbalance (Schmidt et al., 2017) and the global mean temper-
ature (Rohde and Hausfather, 2020). Even though all ensemble members, including the default parameter set, show biases

with respect to observations no member of the nine-member ensemble can clearly be disgarded as unrealistic and we deem the
perturbed parameter ensemble suited to investigate the role of AMOC variability on spatio-temporal temperature variability.



Using these nine parameter sets and starting from long pre-industrial control simulations, we performed nine corresponding 2,000 year long experiments covering the past two millenia forced with time varying volcanic and solar forcings, changes in greenhouse-gas concentrations and changes in the orbital parameters. We limit our analysis to the period before 1850 in order
to exclude the increasingly strong antropogenic warming signal over the last 150 years. All other boundary conditions are pre-industrial (Goosse et al., 2010).

## 2.2 PAGES-2k temperature reconstructions

Using a selection of 210 local temperature sensitive proxies from the PAGES 2k database (Ahmed et al., 2013), Neukom et al. (2019) present six different climate field reconstruction (CFR) methodologies to extend the point-data to a full global
coverage. In our current study we will use these six different temperature field reconstructions to investigate the robustness of reconstructed temperature variability and spatial covariance, and to compare with model results. The CFR methods deployed by Neukom et al. (2019) range from basic proxy composites, to advanced statistical techniques that combine the Ahmed et al. (2013) proxy data-set with physical constraints and forcing information from climate-model simulations.

Here we will provide a short summary of the six different CFR methods of Neukom et al. (2019) because of their importance
for our investigation, further details can be found in the original publication: 1) Composite plus scale (CPS) is an index reconstruction method in which the input proxy data are averaged into composite time series, that are in turn scaled to the mean and standard deviation of the reconstruction target over the calibration period. 2) Principal-component regression (PCR) reduces the dimensions of both the target field and the proxy data using principal-component analysis. In this approach the covariance structure of the temperature grid is based on the instrumental record and assumed to be constant for the whole reconstruction
period. 3) Canonical correlation analysis (CCA) uses singular-value decomposition to separately reduce the dimensions of the instrumental temperatures, the proxy data, and the regression coefficient matrix that describes their relationships. The main assumption is that the first few leading modes of the empirical orthogonal function contain most of the variance in the target climate field and the multi-proxy network. 4) GraphEM uses the theory of Gaussian graphical models to reduce the dimensionality of the problem. 5) Data assimilation (DA) optimally combines proxy data with climate-model states. Here offline
data-assimilation is used. The climate model provides an estimate of the prior that is updated on the basis of the proxy observations and an estimate of the errors in both the observations and the prior. 6) Analogue method (AM) is a method that requires a pool of plausible climate fields for which in this case simulations from the PMIP3 project are used. In this method the spatial structure of temperature is provided by the different climate models, while the temporal evolution is obtained from the information contained in the proxy data. Generally one can say that three out of six CFRs are using observational information to obtain
information on the spatial correlation structure (PCR, CCA, GraphEM), two methods base their spatial correlation structure on climate model output (DA and AM) and the sixth method doesn't use any form of additional spatio-temporal information (CPS).

Following the recommendation by Neukom et al. (2019), we also include the multi-method-mean in our analysis. All the CFR-based temperature reconstructions include an uncertainty estimate by means of a 100-member ensemble. In parts of our
analysis we use the ensemble mean, while in other parts the uncertainty is explicitly taken into account. For comparison,





we also include the original continental-scale temperature time series from Ahmed et al. (2013), to which we will refer as PAGES2013 data.

By employing a single data set (Ahmed et al., 2013) of 210 local temperature sensitive proxies in six different CFR methods, the data set of Neukom et al. (2019) allows for a good description of the uncertainties caused by the CFR methods. How-
ever, other sources of uncertainties are not (directly) sampled (e.g. spatial distribution, temporal extent, seasonality effects and climate response Anchukaitis and Smerdon, 2022). It is for instance important to remember that the spatial distribution is strongly biased towards the mid-latitude of the NH and that the maximum number of 210 records quickly decreases to values below 30 prior to the year 800 CE (Anchukaitis and Smerdon, 2022). The CFR methods that are used to extend the spatially and temporally limited point-data to a full global coverage for the past 2000 years, thus become increasingly important going
further back in time.

## 2.3 CMIP5 last millennium simulations

We will therefore compare the results of the *i*LOVECLIM perturbed-parameter ensemble with results from three CMIP5 last millennium simulations that had all necessary variables available at time of writing on ESGF (Taylor et al., 2012, MRI-CGCM3, GISS-E2-R and MIROC5;) and the 13-members of the last millennium ensemble with CESM (Otto-Bliesner et al., 2016).

## 2.4 Observational data-sets

Three out of six CFR methods use the observational temperature data-set HadCRUT4-GraphEM, a data-set that is based on HadCRUT4 (Morice et al., 2012), but that was infilled using the GraphEM method to obtain a complete global coverage over the calibration period over the period 1850-2000 with a resolution of $5° \times 5°$(Neukom et al., 2019). For further comparison of our results with observations, we used the ERA5 observational data-set (Copernicus Climate Change Service, 2017), covering
the period 1979 to 2019. To remove the antropogenic global warming signal, the time-series are detrended using a second-order polynomial fit. Note that the obtained results are not sensitive to the exact definition of the observational period.

## 2.5 Data processing

On all temperature time series presented here, a butterworth filter was applied that effectively removes all variability on time-scales smaller than 50 years. We tested the impact on our results of the window size of the butterworth filter, and found that
our findings are robust at least within a range of 20-100 years filter window (not shown). Leads and lags at multi-decadal time scales are expected between the response of temperatures at a given location and either temperatures at another location or with the AMOC. We investigate the importance of lead-lag relationships for the resulting correlation factors by allowing leads and lags of maximum 100 years and thus finding the highest possible correlation. These results will be referred to as 'lagged' in the remainder of the manuscript while 'non-lagged' refers to a default lag of zero years. This calculation is generally done per
continental-scale region, except in Figure 3 where it is done for all grid cells. Our definition of the continental-scale regions is shown in Figure A1.



## 3 Results

The *i*LOVECLIM perturbed-parameter ensemble of past two millenia simulations allows us to investigate the effect of the amount of ocean-driven internal climate variability on both continental-scale temperature variability and on spatial temperature covariance. In the following the results for these two aspects will be presented together with a comparison with the results for CMIP5 last millennium simulations, PAGES-2K temperature reconstructions and observational data-sets.

### 3.1 Temperature variability

The *i*LOVECLIM ensemble shows a wide range of temperature evolutions for the 8 continental-scale regions (Figure 1). Substantial differences between the ensemble members are simulated for North America, the Arctic, Asia and Europe. For those regions, the amount of variability varies up to a factor of three over the ensemble, showing the large impact of AMOC variability on continental-scale temperature variability in the NH. For South America, Antarctic, Africa and Australasia, the amount of temperature variability is largely unchanged.

Overall the amount of continental-scale temperature variability in the *i*LOVECLIM ensemble and the range found over all ensemble members compares favourably with the results of the selected CMIP5 large millennium simulations (Figure 1). This is especially true for Antarctica, North America, Asia and Europe. For some regions *i*LOVECLIM underestimates variability (South America, Australasia and to a lesser extent Africa) while for the Arctic *i*LOVECLIM overestimates variability.

When comparing the simulated results with the different CFR-based reconstructions of continental-scale temperature variability (including the original PAGES2013 reconstructions; Figure 1), a complex picture emerges. The range of variability in both the *i*LOVECLIM and CMIP5 model results is in agreement with the reconstructions for some continental-scale regions (Antarctica, South America, North America, Asia and to some degree also Europe and Africa), while in others either the *i*LOVECLIM (Australasia) or the CMIP5 (Arctic) are in better agreement.

The importance of the AMOC changes in driving temperatures as simulated with *i*LOVECLIM differs largely per continental-scale region. We find that for Antarctic, South America and Australasia, the correlation between continental-scale temperature time-series and AMOC time-series is low to modest, ranging from 0.1 to 0.5 over the ensemble (Figure 2). For the other regions the temperature-AMOC correlation is higher and ranges from 0.5 up to 0.9. Moreover, it seems that for all continental-scale regions, ensemble members with little AMOC variability (Figure 3) tend to have smaller temperature-AMOC correlations and those with higher AMOC variability tend to show higher correlations. However, the relationship is far from straighforward and this will become more evident in the following section when we look at the corresponding spatial temperature covariance structure.

### 3.2 Spatial temperature covariance

#### Inter-continental temperature correlations

The amount and characteristics of AMOC variability in a simulation not only impacts temperature variability on a site-by-





site basis, but it also strongly shapes the spatial temperature covariance structure. The different AMOC time-series (Figure
3), show multi-decadal to multi-centennial AMOC variability in all simulations. However, the magnitude ranges from under
3Sv in experiment 1 to up to 10Sv in experiment 7. Moreover, these results show that not only the amplitude of the AMOC
variations is important, but also the dominant frequency. For instance, in ensemble members 3 and 6, we find strong but short-
lived variations in the AMOC that only correlate with temperature variations in the northern North Atlantic. On the contrary,
longer-lived AMOC variations as found in for instance ensemble members 2, 4 and 7, impact temperatures throughout the
Northern Hemisphere. An in-depth discussion on the underlying mechanisms is not the scope of this manuscript, but various
AMOC modes have been described previously for the *i*LOVECLIM model (e.g. Friedrich et al., 2010; Goosse and Renssen,
2004; Kessler et al., 2020; Kim et al., 2021).

Because of the differences in the temperature fingerprint of the AMOC variations between the ensemble members, we also find
differences in the way inter-continental temperature correlations are affected. For a total of 7 out of all 28 possible temperature
correlations between our continental regions, we find a significant ($p < 0.05$) relationship with the amount of AMOC variability
(Figure 4), encompassing combinations of all continental-scale regions except Antarctica and South America. We note that
there can be two different reasons for a non-significant relationship between AMOC variability and a given inter-continental
temperature correlation, namely because both continents are not sufficiently affected by AMOC varibility, or that both conti-
nents are always strongly correlated to AMOC variability, no matter if this AMOC variability is strong or weak. We further
note that all significant relationships are positive relationships, meaning that an increase in AMOC variability leads to a higher
degree of spatial coherency between continental-scale regions. Given that the impact of AMOC variability is mostly limited to
the Northern Hemisphere (Figure 3), one could expect inter-continental temperature correlations between continents on both
hemispheres to decrease with stronger AMOC variability (a negative slope in Figure 4), however, we do not find any such
relationships, neither significant or non-significant.

Combining all inter-continental relationships creates an overview of the degree to which the temperature evolutions for the dif-
ferent continental-scale regions are related to one another (Figure 5, constructed following the approach of PAGES-2k-PMIP3,
2015). Overall we find that the characteristics of AMOC-induced temperature variability impacts the inter-continental tempera-
ture covariance structure. We use two members of the full *i*LOVECLIM perturbed parameter ensemble to illustrate the range of
possible solutions, while the results for all nine members of the *i*LOVECLIM ensemble can be found in Figure A2. Examples
of clear changes in intercontinental correlations are between the Arctic and Africa, Africa and Europe or for instance Asia
and North America. For some other regions, the correlations remain mostly low (Antarctic, South America and Australasia),
while between other regions the correlations are always relatively high (North America with Europe, North America with the
Arctic, and between Europa, Africa and Asia). Whether or not we optimize the correlations by considering possible lead-lag
relationships does not lead to large changes (compare Figure A2 and Figure A3). The four CMIP5 last millennium simulations
that are used for comparison in our analysis, highlight that also for different climate models the strength of the inter-continental
temperature correlations ranges from overall low (MRI-CGCM3) to overall high (MIROC-ESM). Taking the CMIP5 simula-
tions into consideration it appears that the inter-continental temperature correlations that are always low or relatively high over
the *i*LOVECLIM ensemble are not a robust feature of climate models in general.





The inter-continental temperature correlations based on the temperature reconstructions yields widely varying results for the

different CFR methods (Figure 6), a range that is not unlike the results obtained for the climate models. The CFR methods that use climate model input to generate the field reconstructions (see method section) generally show high inter-continental temperature correlations (AM and DA), while the CFR methods that use observational constraints to generate field reconstructions show relatively low inter-continental temperature correlations (CCA, PCR and GraphEM). The CFR method that is used to extend the point-data to a full global coverage thus has a large impact on the resulting spatio-temporal temperature

covariance structure. Not applying a CFR method, but using the original PAGES2013 temperature time-series for the different continental-scale regions (Figure 6) results in lower correlations than those found in the results based on any model or CFR.

The CFRs include an estimate of the uncertainty by means of an 100-member ensemble (Neukom et al., 2019). We find that the inter-continental correlations are impacted by how we take this uncertainty into account. We show this by comparing two different ways to compute the multi-method-mean (Figure 6). One can either calculate the inter-continental correlation for

every individual ensemble member before calculating the ensemble mean inter-continental correlations (Figure 6); or one can calculate the ensemble mean temperature time-series per grid cell and based on that calculate the inter-continental correlations (Figure A4). In the latter approach one averages out some variability before calculating the inter-continental correlations, leading the overall higher multi-method-mean results.

Possibly, the low inter-continental correlations between the original PAGES2013 temperature time-series results from the sub-

sampling of a small number of sites per continental-scale region. We test this using the *i*LOVECLIM ensemble by randomly picking a small number of sites per region and calculate the correlations based on that. We find that depending on the sites that are randomly picked, the inter-continental correlations are at best similar to those based on the full-region data, but can also be much lower (Figure A5). The lower-end results for some perturbed parameter ensemble members approaches the original PAGES2013 based inter-continental correlations (Figure 6).


### Land-sea contrast in temperature variability

Another way to compare reconstructed and simulated spatio-temporal temperature correlations are the differences in temperature variability between land and ocean at a given latitude. Before studying simulated and reconstructed long-term (>50 years) land-sea temperature variability ratios in the past two millenia, we first investigate how well climate models and the

PAGES-based CFR compare with observational data sets for the period 1850-2000 (Figure 7). Note that in contrast with all other analyses presented in this manuscript, for this comparison with more recent observations the time series are too short to use 50-year smoothed data and instead we use annual mean temperature time-series. However, the antropogenic global warming signal is removed by detrending the time-series using a second-order polynomial fit. The *i*LOVECLIM ensemble shows land-sea temperature variability ratios close to one for nearly all latitudes. The exceptions are the high latitudes in both hemi-

spheres at which ocean temperature variability dominates, and the NH mid-latitudes where continental temperature variability is larger. The spread over the *i*LOVECLIM perturbed-parameter ensemble is relatively small. For the mid-to-high latitudes of both hemispheres the CMIP5 simulations and the *i*LOVECLIM are in reasonable agreement, however, for the latitudes roughly between 40°and 10°in both hemispheres, the CMIP5 simulations suggest much more temperature variability over land than



over the oceans, with values up to a ratio of 2 to 3. The two observational data-sets that we show here for validation, ERA5 and

HadCrut4_GraphEM generally show land-sea temperature variability ratios close to unity. This is not unlike the *i*LOVECLIM ensemble, but quite different from the CMIP5 results for the latitudes between 40° and 10°. MIROC-ESM differs from the other models and the obvservational data-sets with land-sea temperature variability ratio values of around two in the low latitudes. It is notable that both observational data-sets are also rather different in many places. One possible cause of these differences could be the fact that both observational products cover a different period in time (1850-2000 versus 1979-2019). However, we

find that this has only a minor impact (Figure A6). The cause for the differences should thus be sought in underlying methodological differences of both observational products.

The PAGES-2K CFR methods also show many latitudes with land-sea temperature variability ratios close to unity (Figure 7), in line with the observations and the *i*LOVECLIM ensemble. However, there is a substantial spread amongst the different CFR methods and there is a large bias in the latitude-band 30°N to 70°N in which the CFR methods show substantially more tem-

perature variability over the continents than over the oceans. Even though both models and most CFR methods are constructed using observational information, this validation shows that this does not guarantee a good agreement for higher order metrics like land-sea temperature variability ratios.

Now we turn again to variability on longer, multi-decadal to multi-centennial, time scales (Figure 8). The *i*LOVECLIM ensemble shows results for the past 2000 years that are largely comparable with the observational period, except for much smaller

land-sea temperature variability ratios for the mid-to-high latitudes of the NH. In that region, the model simulates a large impact of the amount of AMOC variability, resulting in ratios that range between 50% more ocean variability than land variability, up to 250% more ocean variability. The different CMIP5 last millennium simulations are overall in good agreement with each other and they show results that are in line with CMIP5 results for the observational period. The main difference being, in line with the *i*LOVECLIM ensemble, smaller ratios for the high latitudes of both hemispheres, indicating increased ocean temper-

ature variability relative to land variability on these time-scales.

The PAGES CFR-based land-sea temperature variability ratios for the past two millenia are again largely comparable with the CFR-based results for the observational period. The main difference is higher ratios in the DA-method for the mid-latitudes of the SH and higher ratios in both the DA-method and the GraphEM-method for the mid-latitudes of the NH, meaning that in both cases land temperature variability has increased relative to ocean variability. Comparing figures Figure A7 and Figure

A9, it appears that the PAGES CFR-based land-sea temperature variability ratios for the past two millenia are biased high for the mid-latitudes of the NH because they underestimate temperature variability over the mid-latitude oceans.

Because of the similarities for all data-sets between the results for the observational period and the past two millenia, we still face a large discrepancy between the different model results, between models and the reconstructions and between the different reconstruction-based CFR methods (Figure 8). The main differences are ratios larger than unity for the CMIP5 models at all

latitudes equatorward of 40°, where the *i*LOVECLIM ensemble and the PAGES-2K results suggest values close to unity, except for values smaller than one in the reconstruction close to the equator. For the mid-to-high latitudes of the SH the models agree quite well with each other with more variability in the ocean than over land, while the reconstructed ratios are close to unity. The main discrepancy is found in the mid-latitudes of the NH. There the *i*LOVECLIM ensemble shows much more variability





in the ocean and increasingly so with stronger AMOC variability, the CMIP5 results suggest land-sea temperature variability
ratios close to or just below unity, and the CFR-based results show much stronger variability over the continents than over the
oceans.

It is important to note that these land-sea temperature variability ratios only give information on relative differences between
temperature variability over land and oceans. If we look at the individual terms we see, for instance, that the *iLOVECLIM*
ensemble simulates more variability both over the continents and over the ocean for the mid-to-high latitudes of the NH com-
pared to the CMIP5 simulations (Figure A7). On the other hand, the *iLOVECLIM* ensemble simulates too little variability
over both the tropical oceans and continents (Figure A7), leading to land-sea temperature variability ratios in the tropics that
are very comparable to those simulated by the CMIP5 models (close to unity), but for the wrong reasons. The lack of tropical
climate variability in *iLOVECLIM* is a know bias in the model (Goosse et al., 2010).

**4    Discussion  Conclusion**

We have presented a perturbed-parameter ensemble of the *iLOVECLIM* Earth system model that is designed in order to have
a large spread in simulated AMOC behavior in terms of magnitude and frequency of centennial AMOC fluctuations. Com-
bined with the PAGES-2k temperature reconstructions (Neukom et al., 2019) and a selected number of CMIP5 last millennium
simulations, this allows us to discuss the potential importance of AMOC variability in driving centennial-scale temporal and
spatio-temporal temperature variability over the past two millenia.

In the *iLOVECLIM* ensemble, the AMOC plays an important role in driving centennial temporal temperature variability,
however the spatial extent of this impact differs strongly, from a regional northern North Atlantic impact, to a hemispheric
wide impact. Previous work has also shown climate models in which the AMOC is an important driver of centennial climate
variability (e.g. Knight et al., 2005).

Previous studies looking at temperature variability in the time-domain have suggested that climate models simulate too little
variability on multi-decadal and longer time scales compared to proxy-based reconstructions (Laepple and Huybers, 2014;
PAGES-2k-PMIP3, 2015). Comparing the *iLOVECLIM* ensemble and CMIP5 simulations for the past two millenia with the
PAGES-2k continental-scale temperature reconstructions we don't find such a model-data mismatch on the multi-decadal to
multi-centennial time scales. Given the spread in both reconstructed and simulated continental-scale temperature variability, we
find that model and data results either overlap, or that the models suggest slightly more variability. From our results and given
the large uncertainty in reconstructed continental-scale temperature variability, it is not clear if increased AMOC variability
would lead to a better model-data comparison in terms of continental-scale temporal temperature variability.

Increasing the strength of internal modes of climate variability like those related to the AMOC, not only increase temperature
variability at a given site, but it can also change the temperature correlations between different continental-scale regions. We
find that for 7 out of the total of 28 possible inter-continental temperature correlations there is a positive correlation with the





magnitude of AMOC variability. None of the inter-continental temperature correlations show a significantly negative correlation with the magnitude of AMOC variability. Whether or not we correct for possible lead-lag relationships within the system has only a minor impact on these results. This implies that, in line with previous suggestions (PAGES-2k-PMIP3, 2015), en-

hanced internal climate variability, in our case driven by the AMOC, leads to an enhance in the spatio-temporal temperature covariance by increasing the strength of climatic teleconnections between regions. Previous studies have suggested that models simulate too high inter-continental temperature correlations (PAGES-2k-PMIP3, 2015), and our results suggest that enhanced AMOC-related climate variability will not resolve such a model-data discrepancy. Possibly, mechanisms of centennial climate variability that have a more local impact could lead to a decrease in the simulated spatio-temporal temperature covariance

structure, or perhaps a mixture of various large-scale modes of variability. However, comparing the inter-continental temperature correlations based on the *i*LOVECLIM ensemble, the CMIP5 models, the diferent CFR's and the original PAGES2013 reconstructions, we conclude that the spread in the results based on both models and reconstructions is so large that we cannot confirm the previously suggested model-data mismatch in terms of inter-continental temperature correlations.

Our comparison of reconstructed and simulated land-sea ratios in terms of temperature variability revealed large differences: between the *i*LOVECLIM results and the CMIP5 results, between the various CMIP5 simulations and between the CFR's. As a result, we conclude that available temperature reconstructions of the past two millenia are very uncertain and do currently not provide constraints on model results. Part of the reason why the reconstructed land-sea ratios are so uncertain is possibly because they are based on continental temperature proxies, not directly on sea-surface temperature proxies. As a result, the

CFR methods play a large role extrapolating continental atmospheric temperature onto neighboring ocean regions. Acknowledging that the temperature reconstructions reflect atmospheric temperatures over the ocean rather than SSTs did improve the model-data comparison in our analysis, but large biases remain (Figures A7 and A8). One can also compare the simulated SSTs and the CFR-based 'ocean temperatures' with a data-set of actual SST reconstructions over the past two millenia (McGregor et al., 2015). However, we find that the amount of reconstructed local SST variability (McGregor et al., 2015) is roughly an

order of magnitude larger than the variability in either *i*LOVECLIM, CMIP5 or the CFR-based temperature estimates (Figures A10 and A11). Note that the SST reconstructions are a collection of individual records, not a CFR, so this comparison is done on a site-by-site basis. In fact, compared to the large discrepancy with the SST reconstructions, the *i*LOVECLIM and the CFR-based temperature estimates are in much better agreement with each other (Figure A12). The mismatch we find with the SST reconstructions of McGregor et al. (2015) seems relevant because some previous indications of model-data mismatches

in terms of long-term temperature variability were based on similar SST reconstructions (Laepple and Huybers, 2014). Our findings suggest that increasing AMOC-related climate variability does not significantly improve the model-data comparison of multi-decadal to multi-centennial local ocean temperature variability. More in-depth studies into the relationship between reconstructed local SST variability on the one hand, and simulated SSTs and CFR-based reconstructed temperature variability on the other hand is needed to understand and resolve these issues.


CFR-based reconstructions are in reasonable agreement with each other for first order variables like continental-scale temper-



ature time series and variability. However, for higher order metrics like inter-continental temperature correlations or land-sea contrast, the differences between the various CFR methods is substantially increased. The uncertainty in these CFR-based higher order metrices is even larger when taking into account the uncertainty within the individual CFR methods given by the 100 ensemble members (compare for instance the inter-continental temperature correlations in Figure A3 and Figure 6). Neukom et al. (2019) suggest to only use the multi-method mean over the six CFR methods, but doing so it would remain unclear how large the uncertainty of the resulting spatio-temporal temperature reconstructions is, hampering the model-data comparison.

Despite the fact that over 500 reconstructed temperature time series cover the past two millenia with relatively small age uncertainties (Ahmed et al., 2013; PAGES-2k-PMIP3, 2015), uncertainties in the resulting CFR's remain relatively large. This, in combination with the relatively small magnitude multi-decadal to multi-centennial temperature variations on the continental scale (for most regions the standard deviation is below 0.2 K), leads to unfavorable signal-to-noise ratios and continuing difficulty to constrain climate model simulations using temperature reconstructions of the past two millenia.

AMOC variability is often thought to be a prominent player in driving multi-decadal to multi-centennial climate change. Indeed our *i*LOVECLIM perturbed-parameter ensemble shows a large impact of AMOC variability on both continental-scale temperature variability as well as the spatio-temporal temperature correlations between the various continents. However, comparing the *i*LOVECLIM results with the PAGES-2k continental scale temperature reconstructions and a selection of CMIP5 past millennium simulations, reveals that uncertainties in both model results and temperature reconstructions hamper our ability to determine the importance of AMOC variability for the climatic evolution over the past two millenia from large-scale diagnostics as the one applied here. It thus remains unclear which magnitude of AMOC variability would lead to a better agreement between simulated and reconstructed temperatures for the past two millenia.

*Data availability.* The *i*LOVECLIM perturbed physics ensemble climate model output is available through the ZENODO repository under DOI: 10.5281/zenodo.6675429 or using the URL: https://doi.org/10.5281/zenodo.6675429

*Author contributions.* PB designed the study and performed the climate model experiments. PB, HG and DR analyzed the results and wrote the manuscript.

*Competing interests.* The authors declare that no competing interests are present.



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





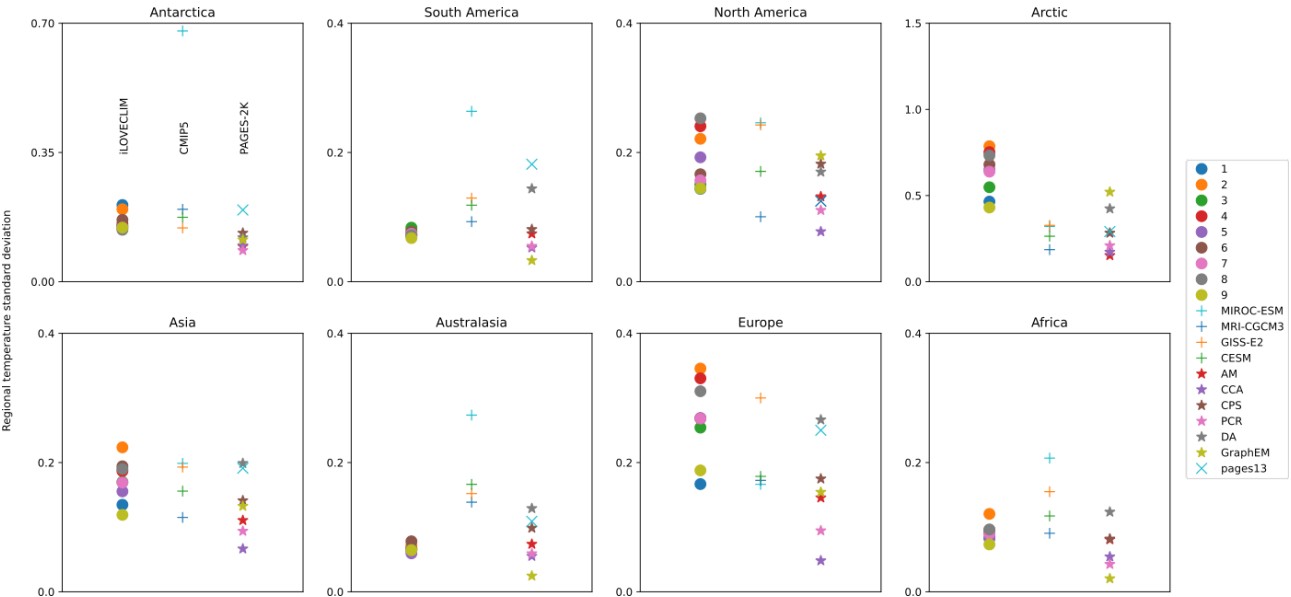

**Figure 1.** Temperature variability (standard deviation in K) for different continental regions. Shown are the nine *i*LOVECLIM ensemble members (left), results for the CMIP5 simulations (middle), ensemble means for the six different CFR-based results from the PAGES-2K data-set (right; Neukom et al., 2019) and the temperature variability based on the original PAGES-2K time-series (Ahmed et al., 2013) for the continental-scale regions for which this data is available. For North America, both the pollen-based and tree-based results are shown. For the CESM results we show the mean of the standard deviation over the ensemble. Note the different y-axis. A 50-year butterworth filter was applied to all results except for the original PAGES-2K time-series which are 30-year averages.



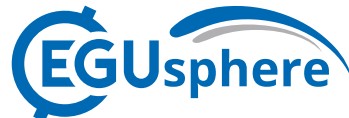

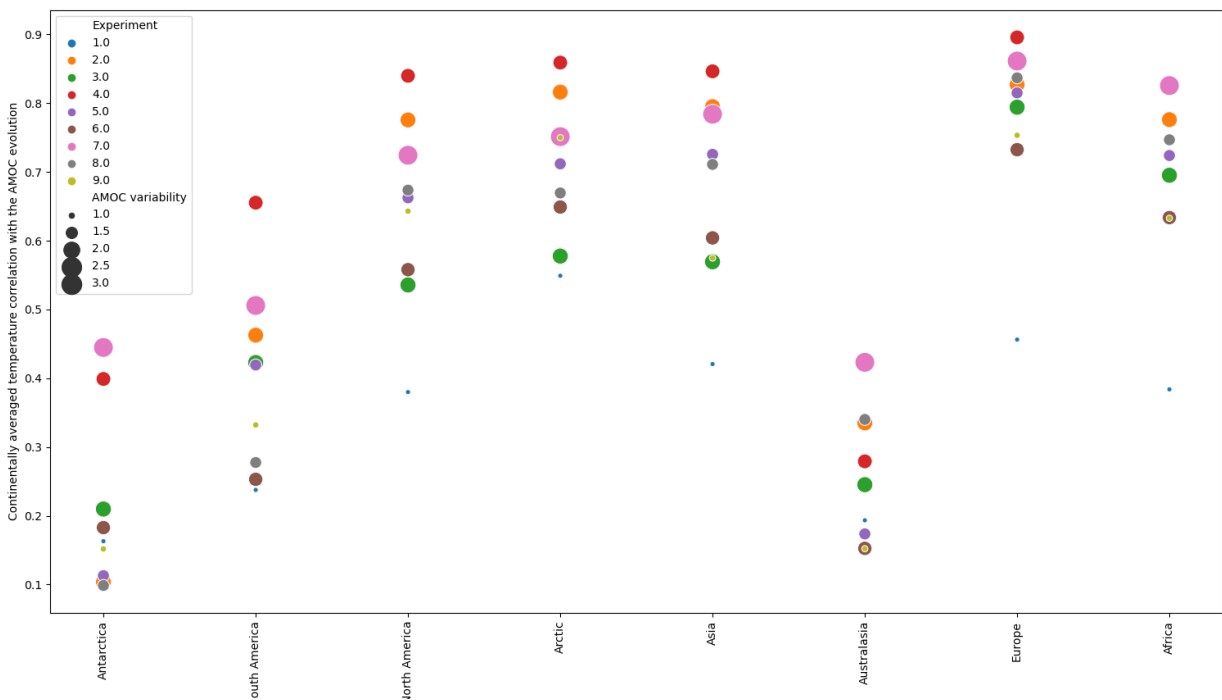

**Figure 2.** Correlation of regional temperature time-series with AMOC evolution in the *i*LOVECLIM ensemble. The different experiments are shown in colors and the amount of AMOC variability (Sv) in the different experiment by the marker size. These results are lagged correlations per continental time serie.



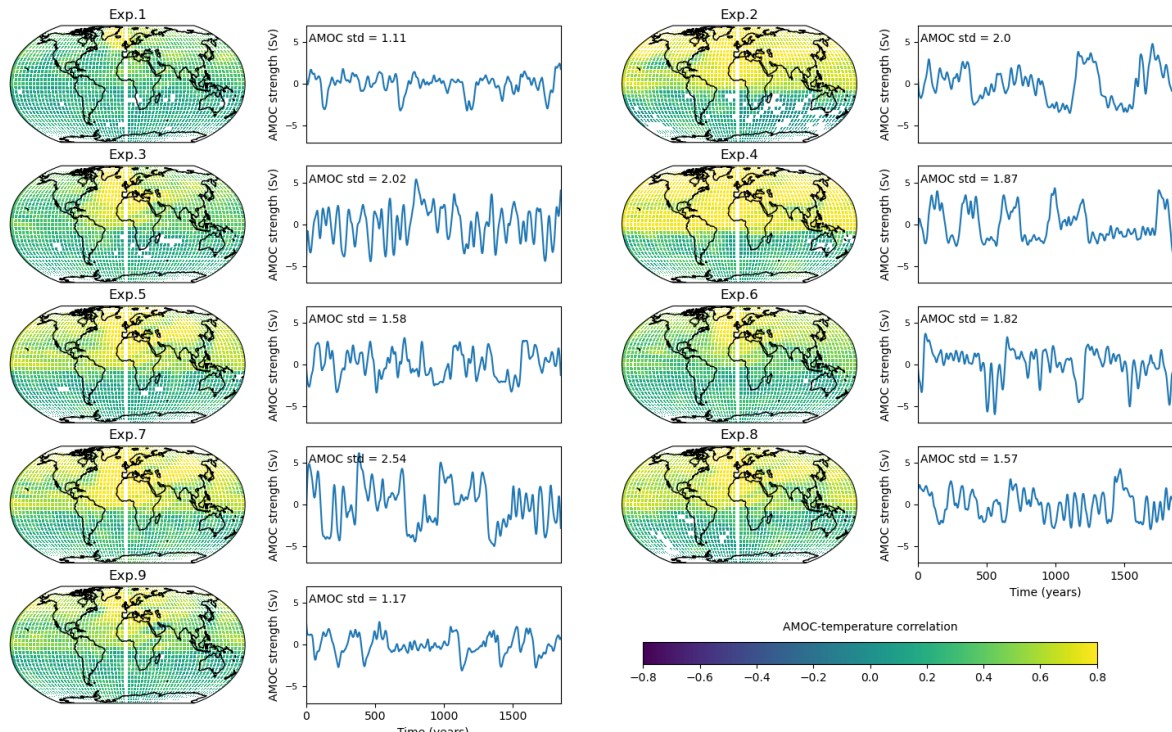

**Figure 3.** For the different ensemble members, the correlation factors (left-hand maps) between grid-based temperature time-series and the AMOC anomalies (maximum overturning stream-function in North Atlantic below 500 m) are shown. These are lagged correlations on a grid-cell basis. Also shown are the time-series of the AMOC (right-hand line-plots with units in Sv) with the AMOC standard deviation in the top-left.




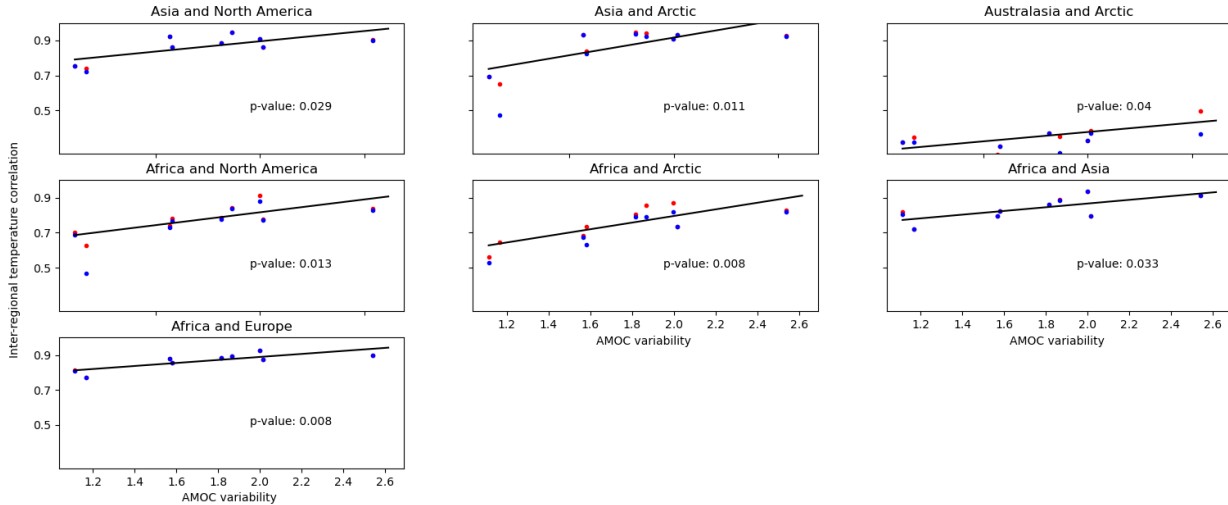

**Figure 4.** Simulated relationship of lagged (red dots) inter-regional temperature correlations as a function of AMOC variability. Results and the linear trend-lines are only shown when the relationship is significant (p<0.05; p-values given the bottom right). For reference, we also show the non-lagged correlation results (blue dots). Calculating the lagged correlations is done per continental time serie.



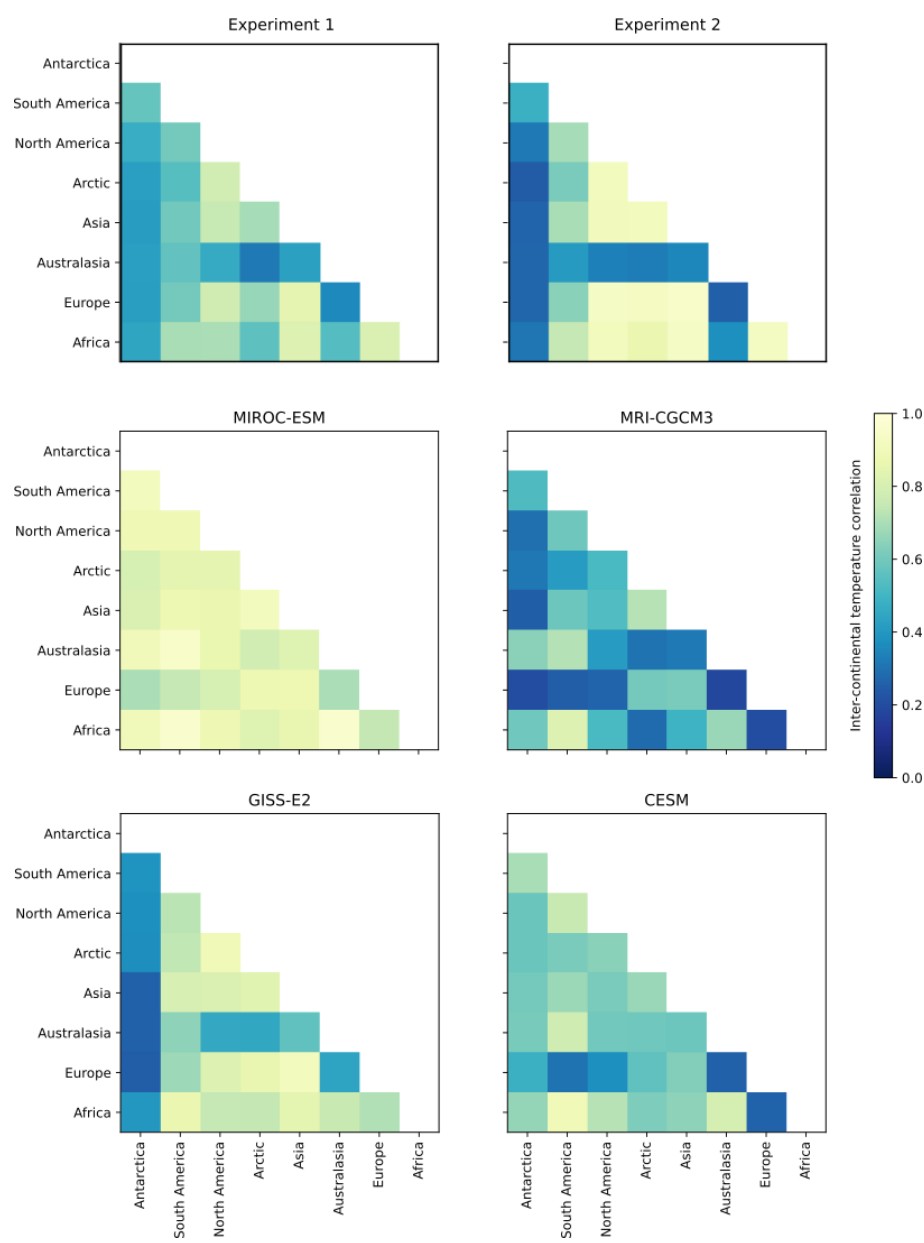

**Figure 5.** Inter-continental temperature correlations for *i*LOVECLIM ensemble members 1 and 2, and for the CMIP5 last millennium simulations. For the CESM results ensemble mean values were first calculated per grid cell before calculating the inter-continental correlations. The *i*LOVECLIM results shown here are lagged correlations. See Figure A2 for the results of all *i*LOVECLIM ensemble members and see Figure A3 for the non-lagged correlations.



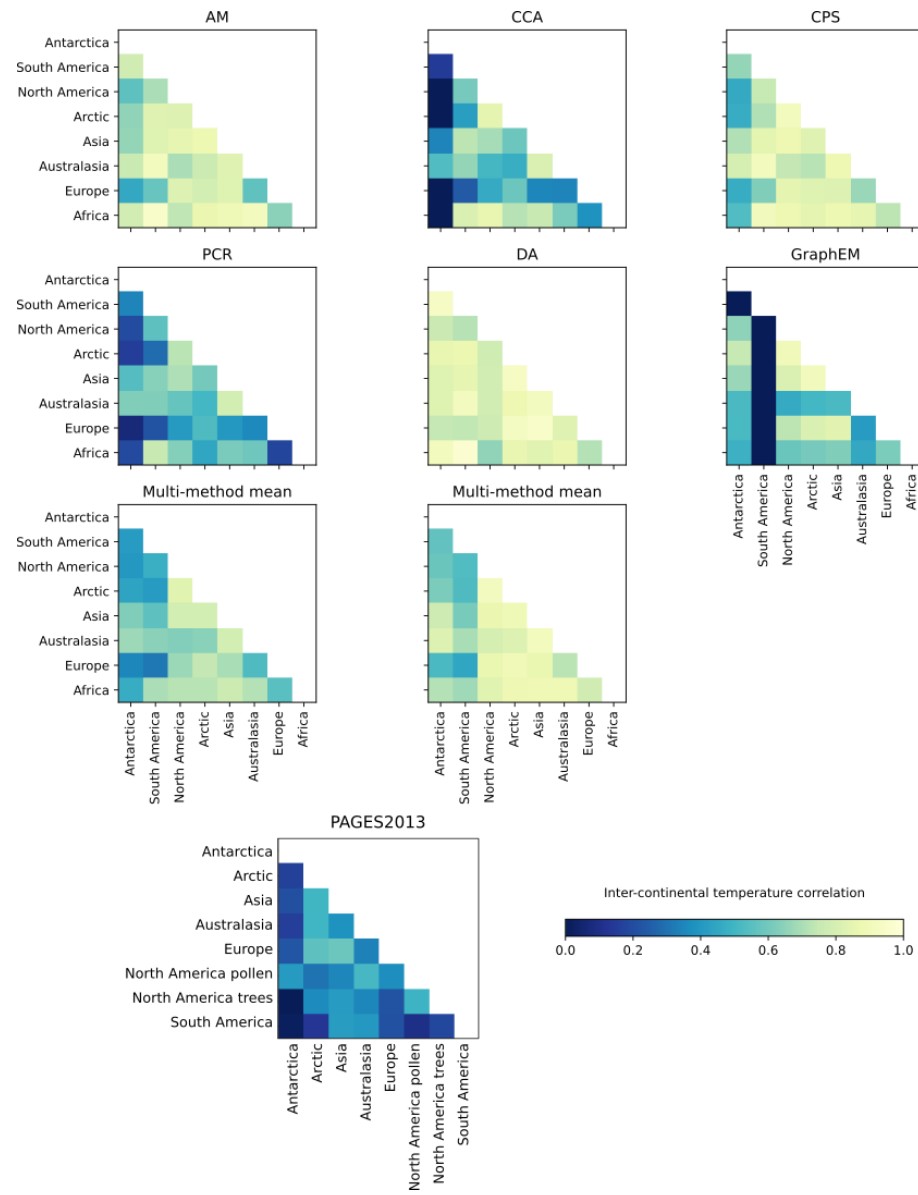

**Figure 6.** Reconstruction based inter-regional temperature correlations. Shown are the PAGES-2K results for the individual CFR methods (Neukom et al., 2019), two different ways to calculate the multi-method-multi-ensemble-mean, and the results based on the original PAGES-2K temperature time series (Ahmed et al., 2013). The top six panels show the results in which individual inter-regional temperature correlations are calculated for every ensemble member, after which the results are averaged for a single CFR. The multi-method-mean of those results is shown on the left of the third row. Another method is to calculate ensemble mean temperature time series for every grid cell, then regional averages and finally calculate inter-regional temperature correlations. The corresponding multi-method-mean is shown here in the middle of the third row while the corresponding results for the individual CFRs are shown in Figure A4. Note that these results are non-lagged correlations and that for the he results based on the original PAGES-2K temperature time series partially different continental-scale regions are shown.




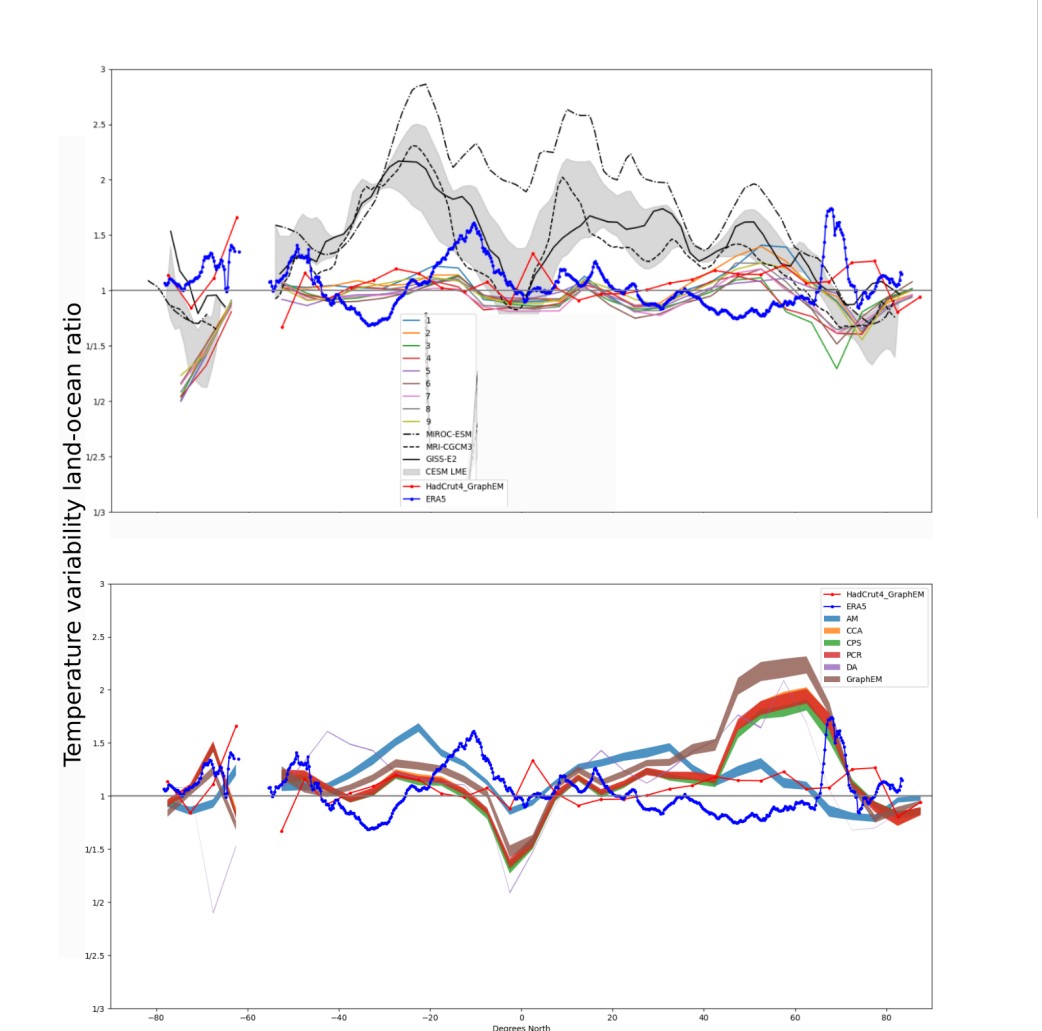

**Figure 7.** Zonally averaged contrast in land-ocean temperature variability as a function of latitude over the observational period (>1850). Atmospheric 2-m temperatures over the oceans are used to calculate ocean temperature variability. Results are shown for the 9-members perturbed-parameter *i*LOVECLIM ensemble (numbers 1 to 9 in top panel), for four different CMIP5 last millennium simulations (top panel) and the results for the six different CFR methods (lower panel). For comparison, in both panels results are shown for the ERA5 and HadCrut4 observational data sets. Note that the ERA5 data-set covers only the period 1979 to 2019. Grey (colored) shading in the top (lower) panel shows the 1-sigma range of all CESM (CFR) ensemble members. Note that the y-axis scale in the right-hand panel is non-linear with for instance 1/1.5 or 1/2 meaning that there is 50% or 100% more variability over the ocean than there is over the continents.



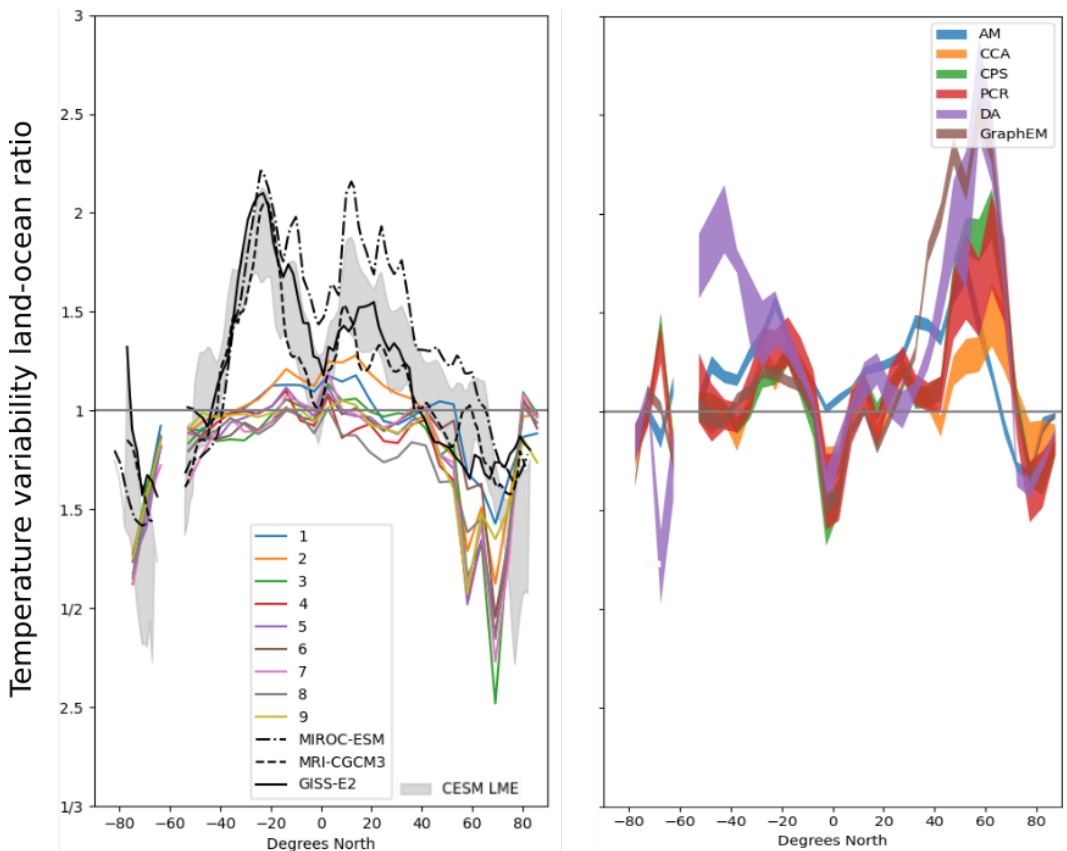

**Figure 8.** Simulated (left) and reconstructed (right) zonally averaged contrast in land-ocean temperature variability as a function of latitude. Atmospheric 2-m temperatures over the oceans are used to calculate simulated ocean temperature variability. In the left-hand panel results are shown for the 9-members perturbed-parameter *i*LOVECLIM ensemble (numbers) and for four different CMIP5 simulations for the last 1000 years. Grey shading shows the 1-sigma range of all CESM ensemble members. The right-hand panel shows for all six different CFR methods the 100-member 1-sigma range. Note that the y-axis scale is non-linear with for instance 1/1.5 or 1/2 meaning that there is 50% or 100% more variability over the ocean than there is over the continents. Add CESM to legend in left-hand panel. The corresponding temperature variability over the continents and over the oceans can be found in Figure A7 (model results) and Figure A9 (reconstructions). The corresponding figure showing the simulated results when using SST's instead of atmospheric 2-m temperatures over the ocean can be found in Figure A8.



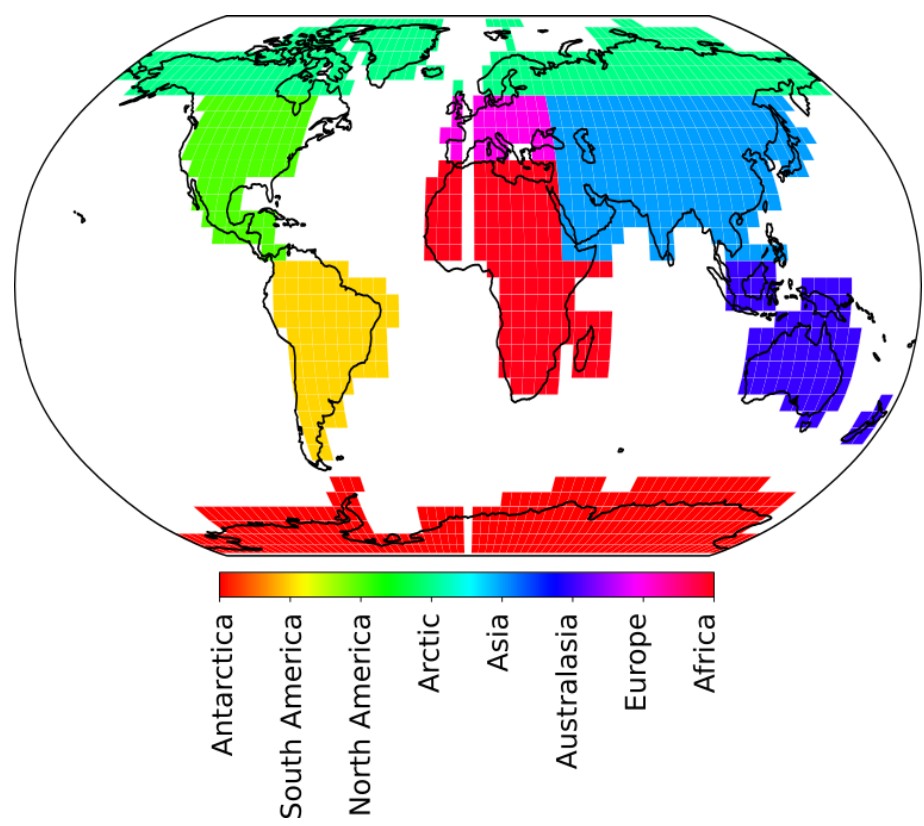

**Figure A1.** Definition of the continental-scale regions that are used in the analysis. Shown here is the *i*LOVECLIM grid.

**Appendix A:  Appendix**





**Figure A2.** Inter-continental temperature correlations for all ensemble members. Top-left is the default simulation. Also indicated per experiment is the magnitude of AMOC variability (standard deviation in Sv) by the size of the red dot in the upper right corner. These values are lagged correlations. See Figure A3 for the results of the non-lagged correlations.







**Figure A3.** Inter-continental temperature correlations for all *iLOVECLIM* ensemble members. Top-left is the default simulation. Also indicated per experiment is the magnitude of AMOC variability (standard deviation in Sv) by the size of the red dot in the upper right corner. In contrast to Figure 5, the values shown here are non-lagged correlations.



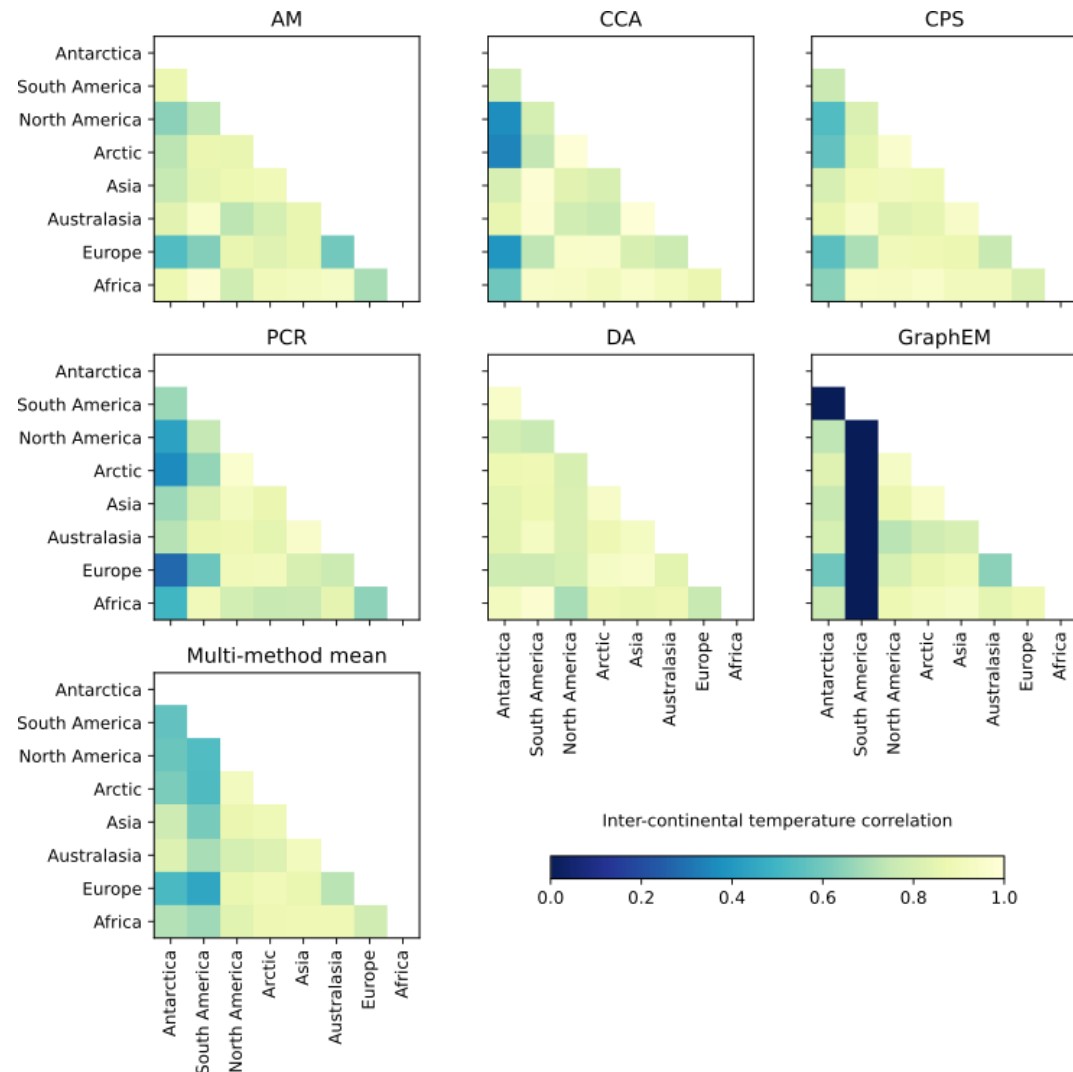

**Figure A4.** PAGES-2K (Neukom et al., 2019) inter-regional temperature correlations for the individual CFR methods. In contrast to Figure 6, ensemble mean temperature time-series per grid cell are used here as the basis to calculate inter-regional temperature correlations for the different CFRs. The multi-method-mean is shown in the lower left. Note that these results are non-lagged correlations.



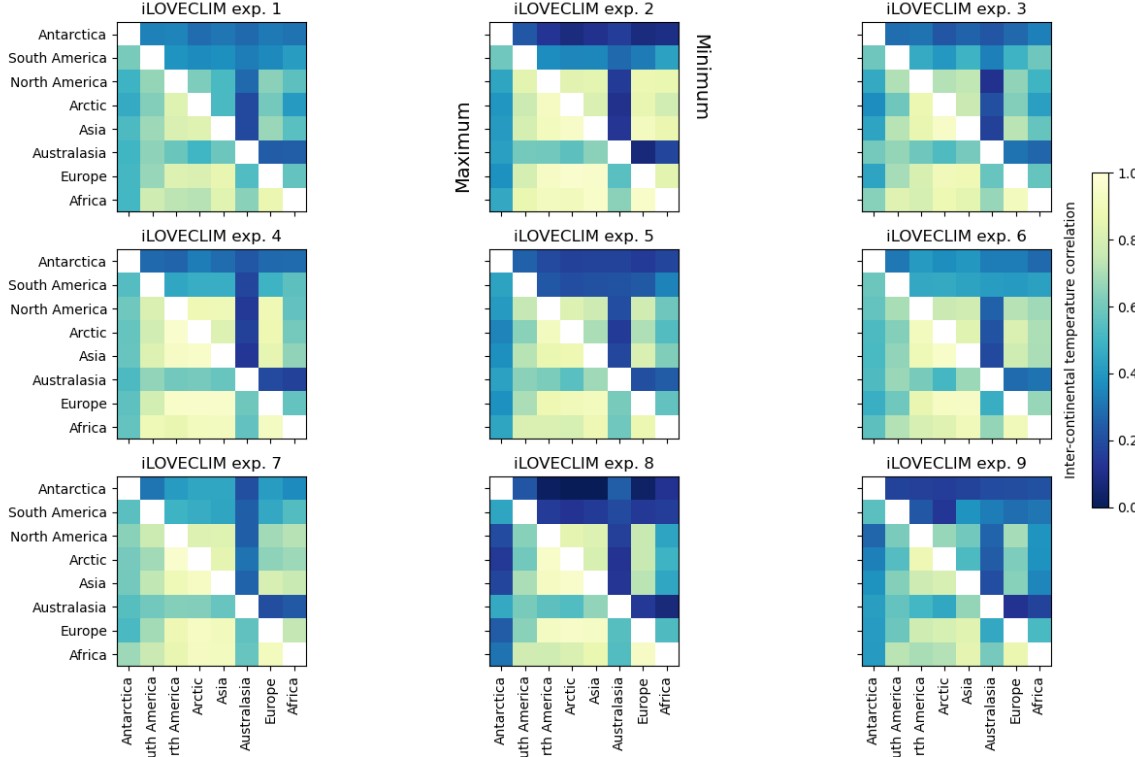

**Figure A5.** Inter-continental temperature correlations for all *i*LOVECLIM ensemble members based on a subsampling of all grid-cells. We randomly pick 10 locations per continental region on which we base the continental-scale average temperature evolution. We do this a total of 30 times and show here the resulting maximum (lower-left corner) and minimum (upper-right corner) inter-continental temperature correlations. Top-left is the default simulation. The values shown here are lagged correlations.



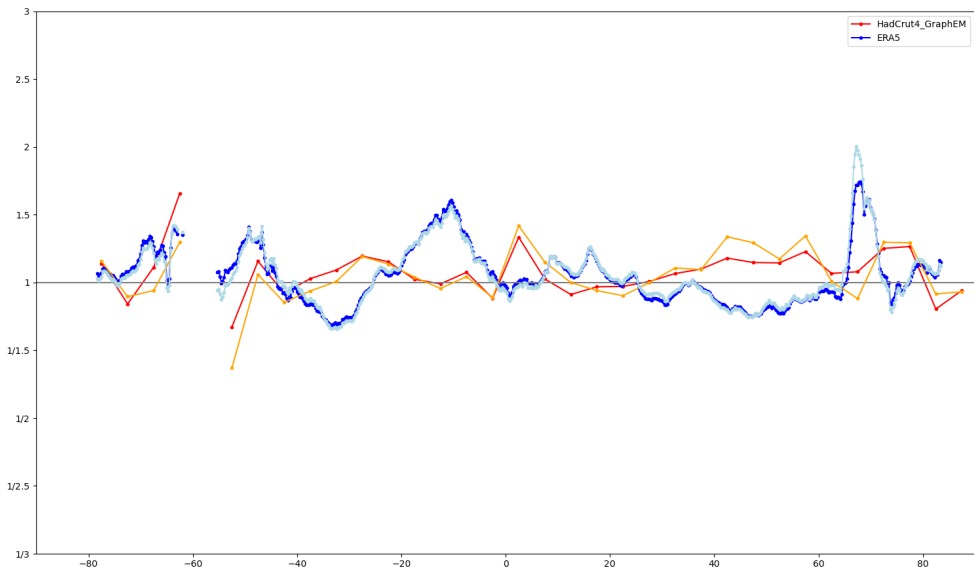

**Figure A6.** Zonally averaged contrast in observed land-ocean temperature variability as a function of latitude for different observational periods. The full observational period for HadCrut4 (red) is 1850-2013 and for ERA5 (dark blue) 1979-2019. Shown for comparison are the result for HadCrut4 (orange) and ERA5 (light blue) when only the period of overlap is used in the calculations (1979-2013). Note that the y-axis scale in the right-hand panel is non-linear with for instance 1/1.5 or 1/2 meaning that there is 50% or 100% more variability over the ocean than there is over the continents.





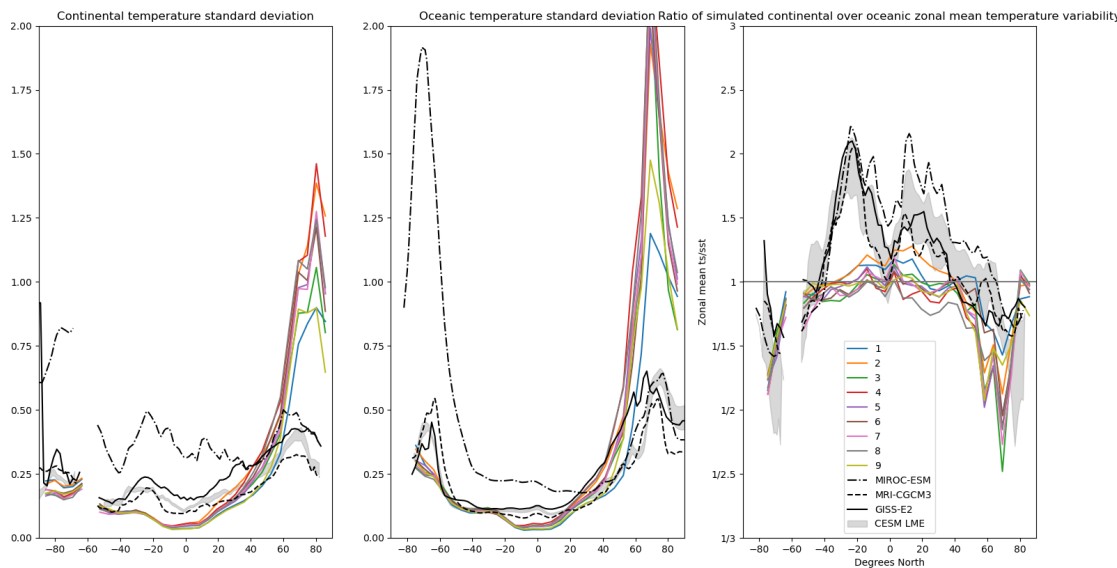

**Figure A7.** Simulated zonally averaged contrast in land-ocean temperature variability as a function of latitude. Depicted are continental variability (left-hand column), ocean variability (middle column) and the ratio of continental over ocean variability (right-hand-column). Atmospheric 2-m temperatures over the oceans are used to calculate simulated ocean temperature variability. Results are shown for the 9-members perturbed-parameter *i*LOVECLIM ensemble (colors) and for three different CMIP5 simulations for the last 1000 years. Grey shading shows the 1-sigma range of all CESM ensemble members. Note that the y-axis scale in the right-hand panel is non-linear with for instance 1/1.5 or 1/2 meaning that there is 50% or 100% more variability over the ocean than there is over the continents.



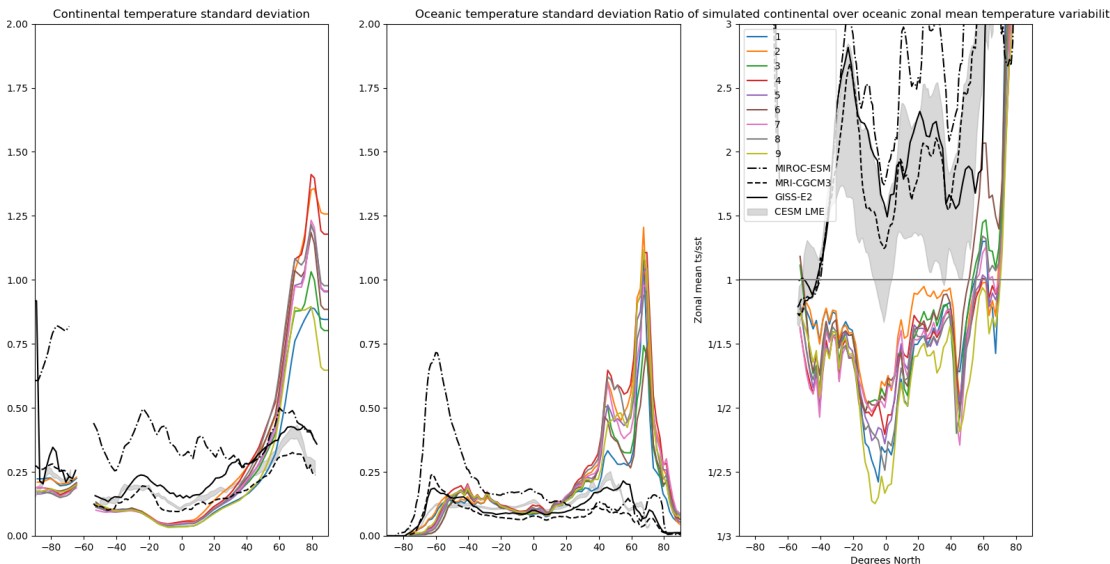

**Figure A8.** Simulated zonally averaged contrast in land-ocean temperature variability as a function of latitude. Depicted are continental variability (left-hand column), ocean variability (middle column) and the ratio of continental over ocean variability (right-hand-column). SSTs are used to calculate ocean temperature variability. Results are shown for the 9-members perturbed-parameter *i*LOVECLIM ensemble (colors) and for three different CMIP5 simulations for the last 1000 years. Grey shading shows the 1-sigma range of all CESM ensemble members. Note that the y-axis scale in the right-hand panel is non-linear with for instance 1/1.5 or 1/2 meaning that there is 50% or 100% more variability over the ocean than there is over the continents.



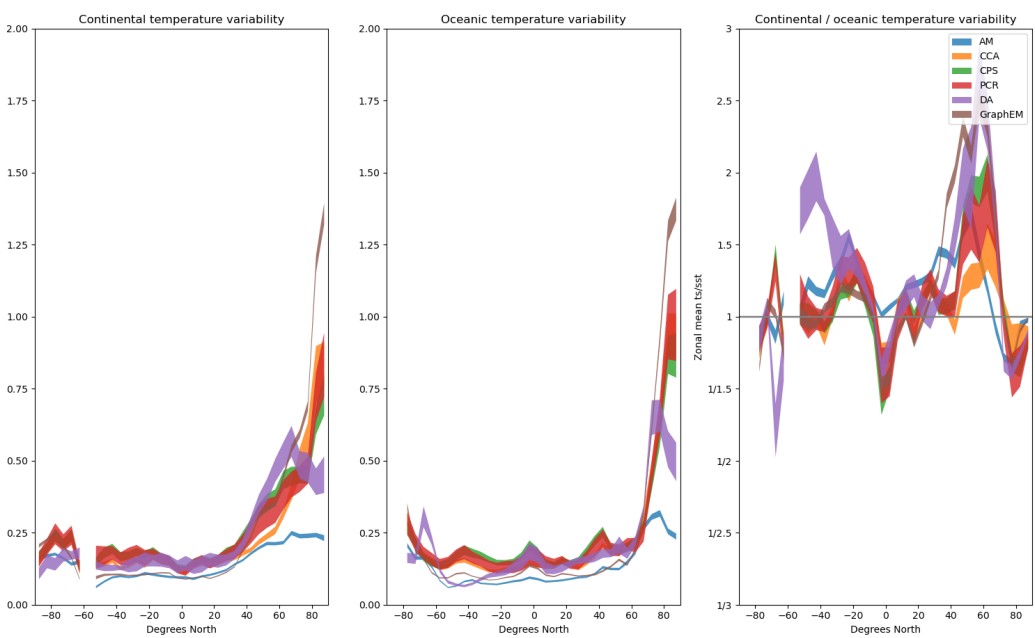

**Figure A9.** Reconstructed zonally averaged contrast in land-ocean temperature variability as a function of latitude. Depicted are continental variability (left-hand column), ocean variability (middle column) and the ratio of continental over ocean variability (right-hand-column). Resuls show for all six different CFR methods the 100-member 1-sigma range. Note that the y-axis scale is non-linear with for instance 1/1.5 or 1/2 meaning that there is 50% or 100% more variability over the ocean than there is over the continents.



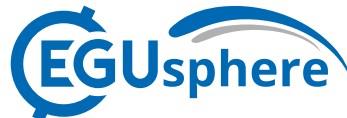

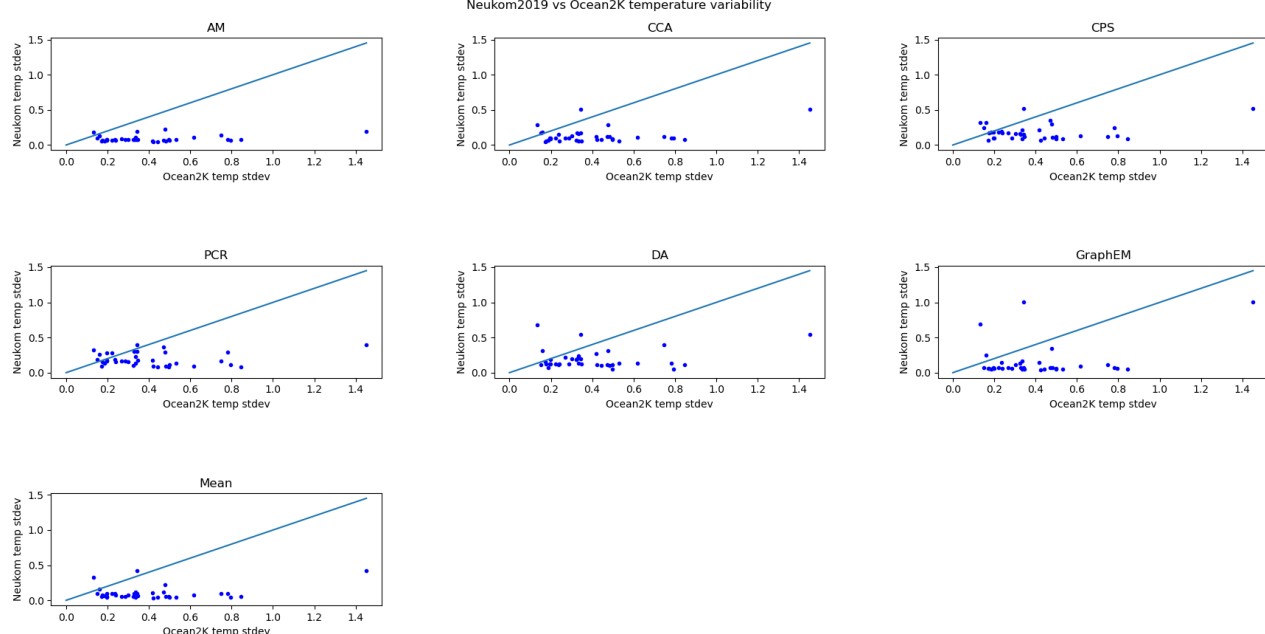

**Figure A10.** Local Ocean2K (McGregor et al. 2015) SST variability versus PAGES-2K data (Neukom et al., 2019) temperature variability at same sites for the different CFR methods.



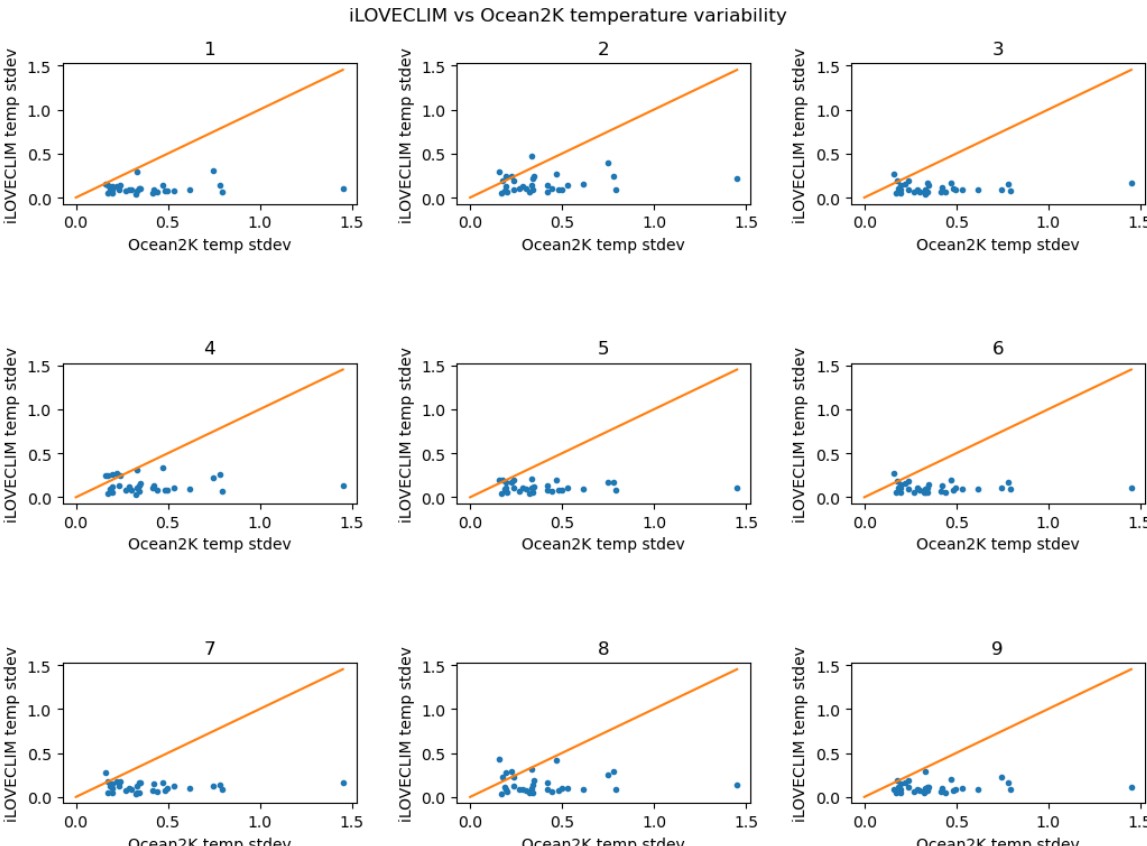

**Figure A11.** Local Ocean2K (McGregor et al. 2015) SST variability versus simulated *i*LOVECLIM SST variability at same sites for the *i*LOVECLIM ensemble members.





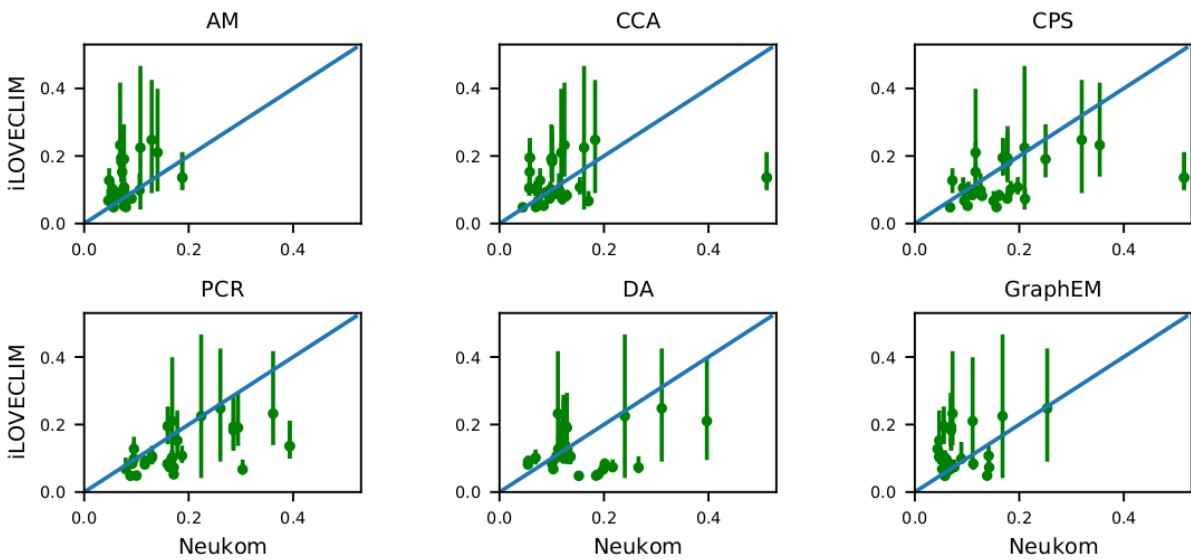

**Figure A12.** Local temperature variability in the PAGES-2K dataset (Neukom et al. 2019) versus simulated *i*LOVECLIM SST variability at Ocean2K locations. For the PAGES-2K dataset the six different CFR methods are shown in the individual panels. The vertical bars in the panels shows the range of simulated *i*LOVECLIM variability over all ensemble members.





**Table A1.** For the individual experiments of the perturbed-parameter ensemble of *i*LOVECLIM, the values of the 10 perturbed parameters are given. Note that experiment 1 is the default. More information on the parameters is given in Table A2.

| Exp. | corAC | corAS | corAN | ampwir | expir | relhmax | evfac | albcoef | albice | avkb |
|------|-------|-------|-------|--------|-------|---------|-------|---------|--------|---------|
| 1 | -0.25 | -0.085 | -0.085 | 1.0 | 0.4 | 0.83 | 1.0 | 0.95 | 0.44 | 1.50E-5 |
| 2 | -0.1 | -0.136 | -0.085 | 1.0 | 0.4 | 0.83 | 1.0 | 0.95 | 0.44 | 1.50E-5 |
| 3 | -0.1 | -0.136 | -0.085 | 0.51 | 0.52 | 0.68 | 0.98 | 0.92 | 0.41 | 8.53E-6 |
| 4 | -0.1 | -0.136 | -0.085 | 0.85 | 0.22 | 0.8 | 0.55 | 0.98 | 0.43 | 8.53E-6 |
| 5 | -0.1 | -0.136 | -0.085 | 0.77 | 0.29 | 0.82 | 0.56 | 0.92 | 0.38 | 1.8E-5 |
| 6 | -0.25 | -0.11 | -0.11 | 0.51 | 0.52 | 0.68 | 0.98 | 0.92 | 0.41 | 8.53E-6 |
| 7 | -0.25 | -0.09 | -0.09 | 0.59 | 0.24 | 0.86 | 0.97 | 1.03 | 0.41 | 1.01E-5 |
| 8 | -0.25 | -0.04 | -0.04 | 0.74 | 0.32 | 0.8 | 0.94 | 1 | 0.42 | 3.97E-6 |
| 9 | -0.25 | -0.09 | -0.09 | 0.55 | 0.41 | 0.88 | 0.51 | 0.94 | 0.4 | 1.1E-6 |



**Table A2.** Meaning of the 10 parameters that are in used to construct the perturbed-parameter ensemble. Also given are the minimum and maximum values that are used in the LHS procedure.

| Parameter | min | max | Description |
|-----------|-----|-----|-------------|
| corAC | - | - | Precipitation correction in the Arctic [Sv]. Water moved from the Arctic to the Pacific. |
| | | | Note: not part of the LHS, fixed values used. |
| corAS | -0.1275 | 0 | Precipitation corrections in the South Atlantic [Sv]. Water moved from the South Atlantic to the Pacific. |
| | | | Note: in experiments 2-5 this parameter is not part of the LHS, instead fixed values are used. |
| corAN | -0.1275 | 0 | Precipitation corrections in the North Atlantic [Sv]. Water moved from the North Atlantic to the Pacific. |
| | | | Note: in experiments 2-5 this parameter is not part of the LHS, instead fixed values are used. |
| ampwir | 0.5 | 1.5 | Scaling coefficient for the longwave radiation scheme |
| expir | 0.2 | 0.6 | Exponent for the longwave radiation scheme |
| relhmax | 0.5 | 0.9 | Precipitation also occurs if the total precipitable water below 500 hPa is above this relevant threshold. |
| evfac | 0.5 | 1 | Maximum evaporation factor over land. |
| albcoef | 0.9 | 1.1 | multiplied factor for the albedo of the ocean in LOVECLIM |
| albice | 0.38 | 0.46 | albedo of sea ice and snow. |
| avkb | 1E-6 | 2.5E-5 | Scaling factor for the minimum vertical diffusion coefficient in the ocean at all depths |