# Peer review of "Internal climate variability and spatial temperature correlations during the past 2000 years"

_EGUsphere, 2022_

## Referee Comment (RC1)

Summary: The study investigates the impact of the Atlantic Overtirnung Circulation on the amplitude and covariance of continental temperatures over the past 1000-2000 years. It is based on comparison of simulations with a climate mode of intermediate complexity (ILoveclim), simulations with a few Earth System Models from the CMIP5 project, additional simulations with the CESM model, and proxy-based temperature reconstructions from PAGES-2K. The initial hypothesis is that the AMOC can impact the level of continental cross-correlation and explain why CMIP5 models tend to overestimate it. The authors conclude that the spread in correlations and amplitude of temperature variability is so large that it is difficult to prove or dismiss this hypothesis.

Recommendation: The manuscript is very well written, the research question is very clearly stated and the study, although in the end not conclusive, is useful on the way to pin down the real physical reason. I have a few comments that the authors may want to consider, but in my opinion, the manuscript can be published with very little changes.

General comment: I enjoyed reading this manuscript. As I wrote, the conclusions are not definitive, but the research questions and the experimental set-up are nice and the study is well conducted. Certainly, this study would be very difficult to conduct with a full fledged Earth System model, so that iLoveclim seems well suited for it.

Particular comments

1) line 136 'CMIP5 last millennium simulations that had all necessary variables bles available at time of writing on ESGF (Taylor et al., 2012, MRI-CGCM3, GISS-E2-R and MIROC5;)'

The sentence may be a bit uprising, as there are other past1000 simulations. Perhaps it would be helpful to state that the critical variable is the AMOC index. Nevertheless, the MPI-ESM-P mode does provide this variable (see attached file).
Probably, it is not worth to repeat the calculations including this model, as the model spread is already large, but if the MPI-ES;-P model could have been included, the sentence should be amended.

2) line 148 a butterworth filter

Butherworth should capitalized as it is the name of the scientist that designed this filter

3) Fig 2 correlations per continental time serie

time series is also singuar

4) line 205  Given that the impact of AMOC variability is mostly limited to the Northern Hemisphere (Figure 3), one could expect inter-continental temperature correlations between continents on both hemispheres to decrease with stronger AMOC variability (a negative slope in Figure 4), however, we do not find any such relationships, neither significant or non-significant.

This expectation assumes that the AMOC does have an impact on the southern hemisphere. If this impact is too low compared to other mechanisms of internal variability and to the impact of the external forcing, the correlations would be already too low to be able to further decrease.

5) line 305 4 Discussion Conclusion

Discussion and Conclusion

6) line 328 ....with the magnitude of AMOC variability. Whether or not we correct for possible lead-lag relationships within the system has only a minor impact on these results.

I think the stop after 'variability' should be replaced by a comma.

7) Perhaps the authors can consider placing the results of Figure 1 and 2 within ecah continent on a global map. Not totally necessary but perhaps more useful

---

## Author Comment (AC1)

Reply to anonymous referee 1

**We thank the referee for the constructive comments and we are happy to hear that the manuscript was enjoyable to read.**
**In the following, the original referee comments are given in italics and the reply by the authors in bold.**

*Summary: The study investigates the impact of the Atlantic Overtirnung Circulation on the amplitude and covariance of continental temperatures over the past 1000-2000 years. It is based on comparison of simulations with a climate mode of intermediate complexity (ILoveclim), simulations with a few Earth System Models from the CMIP5 project, additional simulations with the CESM model, and proxy-based temperature reconstructions from PAGES-2K. The initial hypothesis is that the AMOC can impact the level of continental cross-correlation and explain why CMIP5 models tend to overestimate it. The authors conclude that the spread in correlations and amplitude of temperature variability is so large that it is difficult to prove or dismiss this hypothesis.*

*Recommendation: The manuscript is very well written, the research question is very clearly stated and the study, although in the end not conclusive, is useful on the way to pin down the real physical reason. I have a few comments that the authors may want to consider, but in my opinion, the manuscript can be published with very little changes.*

*General comment: I enjoyed reading this manuscript. As I wrote, the conclusions are not definitive, but the research questions and the experimental set-up are nice and the study is well conducted. Certainly, this study would be very difficult to conduct with a full fledged Earth System model, so that iLoveclim seems well suited for it.*

*Particular comments*
*1) line 136 'CMIP5 last millennium simulations that had all necessary variables bles available at time of writing on ESGF (Taylor et al., 2012, MRI-CGCM3, GISS-E2-R and MIROC5;)'*
*The sentence may be a bit uprising, as there are other past1000 simulations. Perhaps it would be helpful to state that the critical variable is the AMOC index. Nevertheless, the MPI-ESM-P mode does provide this variable (see attached file).*
*Probably, it is not worth to repeat the calculations including this model, as the model spread is already large, but if the MPI-ES;-P model could have been included, the sentence should be amended.*
**We agree with the referee that more CMIP5 last millennium simulation have been performed. It is not our intention to perform a full model inter-comparison of temperature variability in CMIP5 simulations. Our focus is really on the iLOVECLIM ensemble and the PAGES2K reconstructions. The CMIP5 simulations are shown to provide insights into what the results of the full CMIP5 last millennium ensemble might look like. Nonetheless, we agree that our formulation in the manuscript is not clear about this. We therefor rewrote it " For further comparison we also include results from three randomly selected single member CMIP5 last millennium simulations (MRI-CGCM3, GISS-E2-R and MIROC5) and the 13-members of the last millennium ensemble with CESM (Otto-Bliesner et al. 2016). Including these CMIP5 simulations allows us to put the results of the iLOVECLIM perturbed parameter ensemble and the PAGES-2k reconstructions in perspective."**

*2) line 148 a butterworth filter*
*Butherworth should capitalized as it is the name of the scientist that designed this filter*
**We have corrected this throughout the manuscript.**

*3) Fig 2 correlations per continental time serie*
*time series is also singuar*
**We have corrected this throughout the manuscript.**

*4) line 205 Given that the impact of AMOC variability is mostly limited to the Northern Hemisphere (Figure 3), one could expect inter-continental temperature correlations between continents on both hemispheres to decrease with stronger AMOC variability (a negative slope in Figure 4), however, we do not find any such relationships, neither significant or non-significant.*
*This expectation assumes that the AMOC does have an impact on the southern hemisphere. If this impact is too low compared to other mechanisms of internal variability and to the impact of the external forcing, the correlations would be already too low to be able to further decrease.*
**We agree that if correlations between continents in the NH and SH are already very low for a small amount of AMOC variability, than they could not decrease any further anyway regardless of increases in the amount of AMOC variability. However, we do not agree with the general notion that we are expecting that the AMOC is affecting the SH. If a NH continent A sees an increase in temperature variability which is related to an increase in AMOC variability, but a SH continent B is not impacted by this mode of variability, than the temporal correlation of temperatures between continents A and B will decrease.**

*5) line 305 4 Discussion Conclusion*
*Discussion and Conclusion*
**Corrected.**

*6) line 328 ....with the magnitude of AMOC variability. Whether or not we correct for possible lead-lag relationships within the system has only a minor impact on these results.*
*I think the stop after 'variability' should be replaced by a comma.*
**We prefer to keep the scentence as it is. This is because this particular line about the impact of lead-lag relationships, not only applies to the line directly before, but to all three lines before. We think this is more clearly reflected by keeping the full stop.**

*7) Perhaps the authors can consider placing the results of Figure 1 and 2 within ecah continent on a global map. Not totally necessary but perhaps more useful*
**Thanks for this suggestion. Although we agree that this would make the figures visually more attractive, we prefer to keep the figures as they are because placing them on a map will effectively put each of them on their own vertical axis, making it much more difficult to compare the results from the different continents with each other.**

---

## Author Comment (AC2)

Reply to anonymous referee 1

**We thank the referee for the constructive comments.**
**In the following, the original referee comments are given in italics and the reply by the authors in bold.**

*Review of: Internal climate variability and spatial temperature correlations during the past 2000 years*

*I think this paper is in very good form. I only have a few minor comments.*

*Lines 10-12: I found this sentence confusing: 'However, combining the iLOVECLIM results with CMIP5 model results and various PAGES-2K temperature field reconstructions, we find that neither model results or reconstructions are robust.' The models and reconstructions are not robust in what sense? I think this sentence could be more clearly re-worded to be more specific or perhaps even cut entirely given the content that follows it.*
**Thanks for pointing this out. We have simplified the text by combining the two sentences into "However, combining the iLOVECLIM results with CMIP5 model results and various PAGES-2K temperature field reconstructions, we show overall agreement for the magnitude of continental temperature variability in models and reconstructions, but both the simulated and the reconstructed ranges are large."**

*All of the main text figures have quite small font sizes on nearly all the labels and axes. Their readability would be much improved by increasing the font sizes of everything in the figures.*
**Thanks for pointing this out. We updated all main text figures to increase font sizes where possible.**

*Could you comment in the paper on how the spatial resolution of the different datasets may influence the comparisons you've done? LOVECLIM is 3 degrees, the PAGES2k reconstructions are 5 degrees, the CMIP models are ~2 degrees, and the proxy data is based on individual points, so there's a lot of spatial averaging differences that could affect the comparisons.*
**There seem to be two aspects to this very relevant question. Firstly, we did not find any evidence to suggest that the differences in spatial resolution between the different products that we use (2-5 degrees) impacts our findings. First averaging all fields to a common 5 degree resolution yields the same results.**
**There are, however, important questions related to the general comparison of point-based temperature reconstructions (like McGregor et al. 2015) and coarse resolution gridded products like model output or CFR-based reconstructions. We partially addressed this question in the discussion on lines 357 to 359 of the original manuscript. However, since this is an important question, we have rephrased it on lines 355-358 to read "More in-depth studies are need to understand and resolve the differences between reconstructed point-based temperature variability on the one hand, and temperature variability in coarse resolution products (2-5\ degree{} in this study), like model results and CFR-based reconstructions, on the other hand."**

---

## Author Response (AR3)

Reply to second review by reviewer 2:

I thank the authors for considering my comments. I think the paper is ready for publication.

I have just one minor suggestion on my point 6 of the previous review:
6) line 328 ....with the magnitude of AMOC variability. Whether or not we correct for possible lead-lag relationships within the system has only a minor impact on these results.
I think the stop after 'variability' should be replaced by a comma.
We prefer to keep the scentence as it is. This is because this particular line about the impact of lead-lag relationships, not only applies to the line directly before, but to all three lines before. We think this is more clearly reflected by keeping the full stop.

I think he sentence is even in that case not formulated in proper English, but I guess that in that case the editing services will pick it up. I would rather write ...' This happens whether or not we correct.…'

**We thank the reviewer for pointing this out and we have changed it accordingly.**